# Scalable lipid droplet microarray fabrication, validation, and screening

Tracey N. Bell[1☯], Aubrey E. Kusi-Appiah[1☯¤], Vincent Tocci[1], Pengfei Lyu[2], Lei Zhu[3], Fanxiu Zhu[4], David Van Winkle[5], Hongyuan Cao[2], Mandip S. Singh[6], Steven Lenhert[1]*

1 Department of Biological Science and Integrative NanoScience Institute, Florida State University, Tallahassee, Florida, United States of America, 2 Department of Statistics, Florida State University, Tallahassee, Florida, United States of America, 3 Department of Chemistry & Biochemistry, Florida State University, Tallahassee, Florida, United States of America, 4 Department of Biological Science, Florida State University, Tallahassee, Florida, United States of America, 5 Department of Physics, Florida State University, Tallahassee, Florida, United States of America, 6 College of Pharmacy and Pharmaceutical Science, Florida A&M University, Tallahassee, Florida, United States of America

☯ These authors contributed equally to this work.
¤ Current address: Viro Research, Snellville, Georgia, United States of America
* lenhert@bio.fsu.edu

**Data Availability Statement:** All relevant data are within the manuscript.

**Funding:** The paper was partially supported by funding received by SL from the National Institute of General Medical Sciences, grant number R01

## Abstract

High throughput screening of small molecules and natural products is costly, requiring significant amounts of time, reagents, and operating space. Although microarrays have proven effective in the miniaturization of screening for certain biochemical assays, such as nucleic acid hybridization or antibody binding, they are not widely used for drug discovery in cell culture due to the need for cells to internalize lipophilic drug candidates. Lipid droplet microarrays are a promising solution to this problem as they are capable of delivering lipophilic drugs to cells at dosages comparable to solution delivery. However, the scalablility of the array fabrication, assay validation, and screening steps has limited the utility of this approach. Here we take several new steps to scale up the process for lipid droplet array fabrication, assay validation in cell culture, and drug screening. A nanointaglio printing process has been adapted for use with a printing press. The arrays are stabilized for immersion into aqueous solution using a vapor coating process. In addition to delivery of lipophilic compounds, we found that we are also able to encapsulate and deliver a water-soluble compound in this way. The arrays can be functionalized by extracellular matrix proteins such as collagen prior to cell culture as the mechanism for uptake is based on direct contact with the lipid delivery vehicles rather than diffusion of the drug out of the microarray spots. We demonstrate this method for delivery to 3 different cell types and the screening of 92 natural product extracts on a microarray covering an area of less than 0.1 cm². The arrays are suitable for miniaturized screening, for instance in high biosafety level facilities where space is limited and for applications where cell numbers are limited, such as in functional precision medicine.

GM107172. The funders had no role in study
design, data collection and analysis, decision to
publish, or preparation of the manuscript.

## Introduction

Cell culture is widely used for biological research, production of biological materials such as
antibodies, tissue engineering, and drug discovery. Exposure of cells to a stimulus followed by
measurement of a response is the basis of nearly all cell culture experiments. In the case of
drug discovery and development, drug candidates are typically dissolved in the solution in
which the cells are grown, and the response of the cells to the drug candidates is monitored. In
order to maximize the throughput or number of experiments carried out over time, robotic
fluid handling systems are widely used with 96, 384, or 1536 well plates at the cell culture site
[1–3]. In addition to pipetting drug candidate libraries of more than 100,000 compounds, high
throughput screening also requires pipetting of cells, media, and assay reagents in a sterile and
precisely controlled biocompatible environment.

Microarrays are a promising solution to increasing the throughput of cell culture screening
by removing the need for fluid handling of the drugs from site of the cell culture, which would
allow mass production and distribution. The remaining cell culture steps can then be carried
out in larger wells, with single pipetting events. Screening of hundreds-of-thousands of drug
candidates (i.e., high throughput screening) requires costly and cumbersome fluid handling to
add cells, reagents, and drug candidates which limits the rate of drug discovery. With drugs
needing to be arrayed at the site of cell culture each time, each lab needs a robot. Mass produc-
tion of arrays at a manufacturing location with a single robot for distribution saves time and
space. To miniaturize high throughput screening, small compartments would allow drugs to
be screened close together without contaminating each other. Lipids naturally form small com-
partments and therefore are ideal to solve the technological problem.

Microarrays are particularly promising for applications where reagents or space are limited,
such as in the screening of natural products, high biosafety level facilities, and in the ex vivo
testing of drugs on sample from patient biopsies. Natural products can be difficult to obtain in
large quantities. Despite this challenge, approximately 50% of all approved small molecule
drugs between 1981 and 2019 have been natural products, natural product derivatives or ana-
logues even though only a negligible fraction of high throughput screening interrogations have
used natural products [4]. Furthermore, many lipophilic natural products may be missed due
to the need for delivery through aqueous solutions. Fluid handling is especially limiting for
screening drugs in biosafety level (BSL) 3/4 containment. Fluid handling equipment that con-
tacts a BSL-3/4 pathogen must be decontaminated before being removed from containment.
In the case of fluid handling robots that cannot be autoclaved or exposed to corrosive chemical
decontamination, the equipment needs to be disposable or committed to the BSL-3/4 facility
[5]. A third application where reagents are limited is in functional precision medicine. Func-
tional precision medicine is based on the idea that cells taken from patient's biopsy can be
exposed to drugs ex vivo in order to determine optimal responses [6]. Assays such as BH3 pro-
filing have proven effective in determining efficacy in a variety of different cancers [7–11].
High throughput screening has found several small molecules capable of differentiating stem
cells [12]. A challenge lies in obtaining enough fresh primary cells to test different therapeutics
and/or combinations [13]. One approach is to grow patient derived cells [7–11]. However, it is
known that as cells proliferate in culture, adaptation of cells to the artificial environment often
results in the cells not responding to drugs as they would in the organism [14]. A point of care
microarray that could assay the cultured cells quickly after the biopsy, but before they have
proliferated and adapted to the cell culture environment is an approach to obtaining more
clinically relevant information about which drugs are more likely to be effective on an individ-
ual patient.

Microarray technologies are an established solution to certain fluid handling problems, such as DNA and protein binding assays [15–21]. For instance, DNA hybridization has been scaled up to allow for thousands of hybridization experiments to be carried out on a single surface. Prior to DNA microarray technology, DNA hybridization experiments were carried out by taking variation of DNA strands, labelling them, then heating the mixtures and allowing them to cool and rehybridize [22]. More samples were able to be run by spotting DNA probes then rotating the membrane to get samples to hybridize [23]. Following that, "dot blot hybridizations" used 48 or 96 samples of bulk DNA to determine a relative quantity of target DNA [24]. Pin spotting is a process that has been developed for a variety of applications that require integration of multiple materials onto the same surface, such as DNA, protein, and polysaccharide microarrays [25–28]. 40,000 different materials can be integrated onto a typical glass slide, overcoming the challenge of individual drug delivery [27–30]. Miniaturization of cell culture has proven more challenging due to difficulties in delivering dosages of different drugs to different cells from a microarray. For instance, covalently linking drugs to surfaces makes it impossible for cells to internalize them [25,31,32]. Drug eluting microarrays have been demonstrated for water-soluble drugs [33–36]. In that approach, compounds are encapsulated into a polymer matrix and arrayed onto a surface prior to cell culture. The cellular response to each drug candidate is assayed at each position on the microarray. A challenge with this approach has been the ability to control the dosage, especially in the case of hydrophobic compounds that do not diffuse through water. Furthermore, drug-eluting microarrays are not waterproof, meaning that pre-treatment with coatings such as collagen and washing causes the drugs to diffuse out of the arrays [20]. These limitations have prevented microarray solutions to high throughput screening problems.

Lipid droplet microarrays provide a potential solution by noncovalently attaching lipophilic materials to a surface through encapsulation into lipid droplets [20,37–40]. Contamination resulting from drug leakage prior to or during cell culture can be avoided with this method as the drugs are not eluted but rather are only taken up by cells upon direct contact with the arrays [39,41–43], allowing quantitative dosages similar to solution delivery to be obtained [44]. Lipid droplets of sub-cellular size were found necessary for cell adhesion to the substrates and dosage control [45]. Sub-cellular lipid droplet microarrays have been fabricated using dip-pen nanolithography and polymer pen lithography [46–48]. Here we use a nanointaglio process due to its potential scalability using printing processes [45,49].

Several aspects of the microarray screening process used here have been described previously (such as pin spotting to create a palette, compatibility with cell culture, delivery of drugs, and quantifying dose response) and are combined for scalability and portability (Fig 1) [38,40,44,45,49]. The process starts with traditional plating by fluid handling in which drug candidates are dissolved in DMSO in 384 well plates. Many libraries are commercially available in this format [50]. Solutions of drug infused lipids are arrayed onto a palette using pin-spotting technology. Typical spot dimensions are 200 μm across spaced 200 μm apart. Spot inhomogeneities likely come from the microarray process and may be improved using inkjet printing [3,38]. The palettes are used to ink stamps for nanointaglio printing to allow for cell adhesion and controlled dosage to adherent cells, and quality control carried out prior to cell culture [39,44,46]. Lipid droplet microarrays require smaller amounts of drugs (nanograms per spot) than solution delivery meaning a single library can be used to make multiple arrays for testing depending on stamp geometry [44,45]. This process is potentially scalable to allow 46,464 tests, or well-equivalents on the area of a standard microtiter plate (Fig 1b–1e).

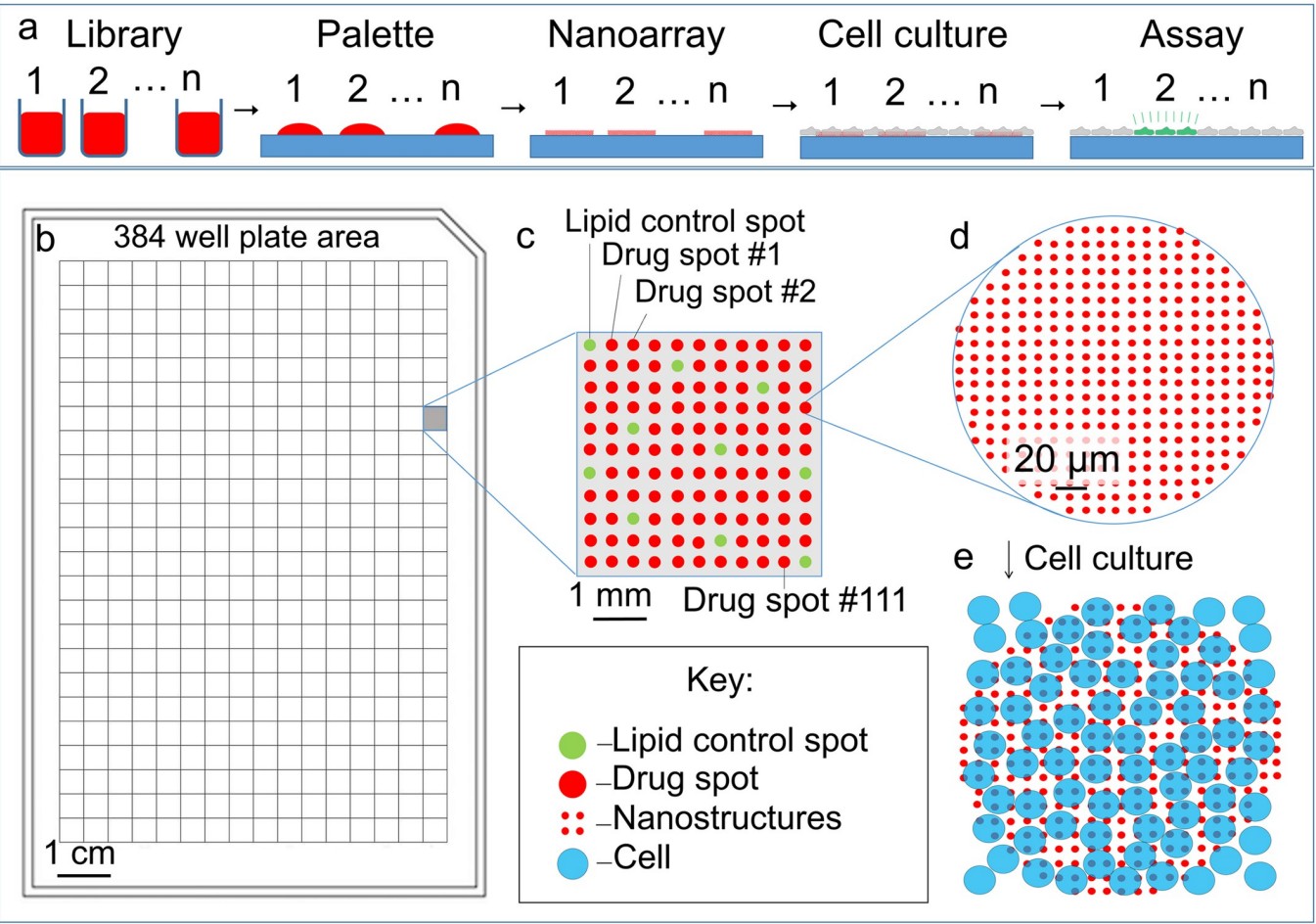

**Fig 1. Schematic illustration of the lipid droplet microarray process.** (a) A library of therapeutic candidates is dissolved in a solvent. The different candidates are arrayed onto a two-dimensional palette by pin spotting. The palette is used to ink a stamp suitable for forming nanoarrays, which are compatible with cell culture. Cells are cultured on the arrays. Assays are then conducted and used to identify effective therapeutics. (b) Example of a 384 well plate. (c) Millimetre scale of 1 well holding over 100 drugs. (d) Micrometre scale showing the subcellular pattern of individual drug subcellular nanostructures. (e) Cell culture over the microarray.

## Results and discussion

### Scalable array fabrication and quality control

Nanointaglio allows for scalable fabrication of the arrays by using a vertical stamping mechanism to create multiple prints from a single pin spotted microarray palette (Fig 2). Larger droplets of several hundred-micrometer diameters on the palette are prepared by standard microarray procedures, with each spot containing a different drug, as indicated by the ink color in Fig 2a. These microarrayed spots are broken into smaller droplets in the nanointaglio process, where multiple droplets of sub-cellular dimensions contain the same drug. Oil-based inks are doped with the fluorescent dye rhodamine-PE for characterization by fluorescence microscopy (Fig 2b–2d). PDMS was used as the stamp material and glass slides used as the substrates. We measured the advancing and receding contact angles on flat PDMS surfaces to be 82 +/- 4° and 77 +/- 4°, respectively, and advancing and receding contact angles measured on the glass substrates were 61° +/- 4° and 42° +/- 2°, respectively. We have previously shown

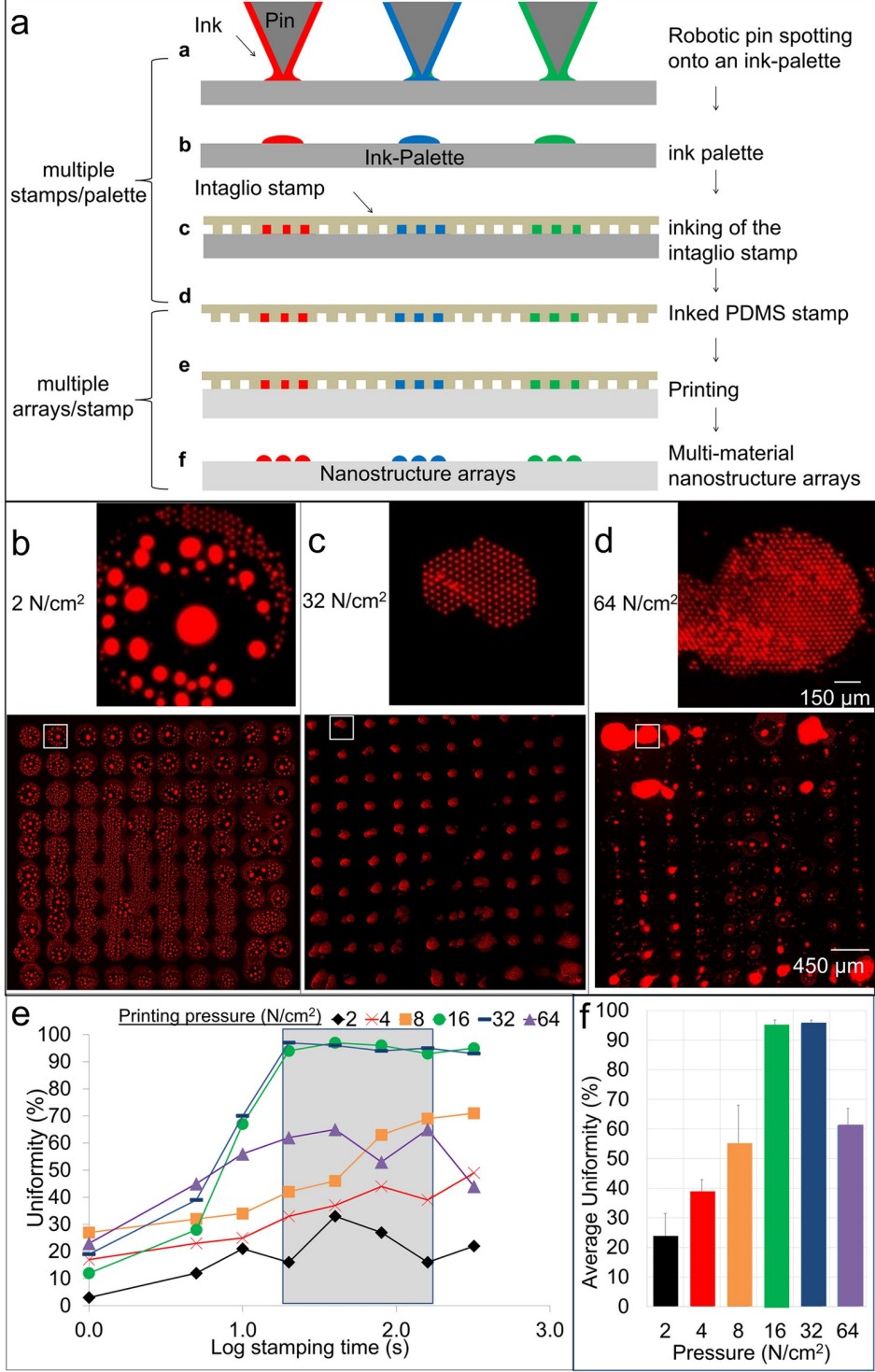

**Fig 2. Scalable fabrication of arrays.** Fabrication and characterization of nanointaglio microarray using vertical stamping mechanism. (a) A schematic showing the stamping process with the option of using multiple inks, allowing us to scale up by making reproducible copies. (b), (c) and (d) are fluorescence images of arrays with a magnified view of the specific location boxed in white of hexanoic acid / castor oil (hex/cas) droplets stamped using vertical pressures of 2, 32, and 64 N/cm2, respectively. Oils were doped with rhodamine-PE for visualization. e) A graph showing the visible individual subcellular nanostructures percentages vs the log of the stamping time in hundreds of seconds the area in grey is the time period of 130–200 seconds. f) A bar graph showing the vertical pressure of 16 to 32 N/cm2 had the highest average uniformity percentage over the span of 130–200 seconds.

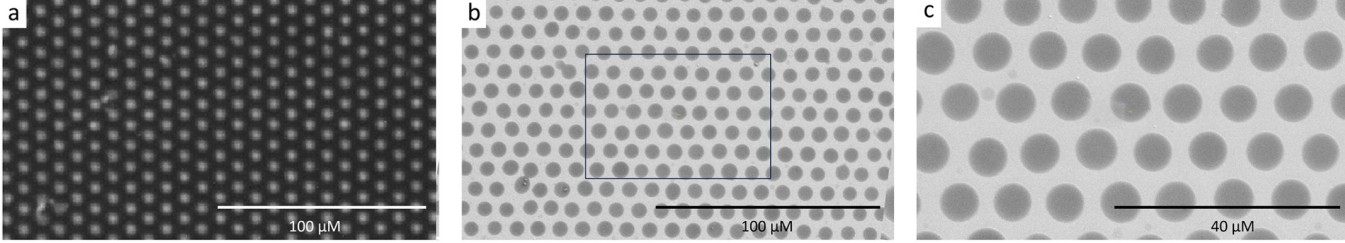

**Fig 3. Micrographs of hexanoic acid / castor oil (hex/cas) droplet arrays prepared in the optimal pressure range.** (a) Representative fluorescence image (b) and (c) are scanning electron microscopy (SEM) images of arrays with a magnified view of the specific location boxed in white of hexanoic acid / castor oil (hex/cas) droplets stamped using vertical pressure of approximately 30 N/cm². Oils were doped with rhodamine-PE for visualization.

the material integration capabilities by printing spots with two different fluorophores [45]. We have also demonstrated a linear the relationship between the multilayer thickness or heights of our subcellular nanostructures measured by atomic force microscopy (AFM) and the intensity of fluorescence emission they produce [46]. This calibration has allowed us to determine droplet volumes and therefore quantify dosage in terms of fluorescence intensity [44]. Here we use fluorescence microscopy to determine an adequate stamping time and printing pressure to acquire repeatable sets of evenly spaced subcellular nanostructures using fluorescent microscopy. In order to quantify this property, we define the print fidelity in terms of a "uniformity" or the percentage of patterned area covered with ink from the stamp that has transferred individual subcellular nanostructures (~5 μm) each to the surface, with dosage being quantified after printing [43–45]. Printing pressures of 16–32 N/cm² and stamp times between 130 and 200 seconds were found to result in optimal uniformity > 95% (Fig 2e–2f). Fluorescence and scanning electron micrographs of arrays printed under optimal conditions are shown in Fig 3. Spots of different compounds would have diameters of 200 μm and a pitch of 400 μm, therefore a spot density of 625 spots per cm² can be achieved. With printing times on the order of one minute per print, this would allow production of arrays at a throughput of 70 million different compounds per day in a roll-to-roll compatible process [51]. Compound libraries are routinely screened many times, often in different labs as well as in the same lab on different assays, for instance with different cell lines and pathogen strains. Using roll-to-roll fabrication would allow multiple copies for different labs to test the same compounds, experimental variation, or different assays, with different cell types and strains. Currently, libraries are copied into plates, shipped, and tested at multiple facilities. For example, the compounds in the National Center for Advancing Translational Sciences (NCATS) Pharmaceutical Collection (NPC) has been used to screen multiple concentrations, repeated trials, multiple cell lines, follow up studies, etc.[52–56] Nearly 200 peer review articles are currently available that used the NPC [57,58].

Although polydimethylsiloxane (PDMS) is convenient for prototype microcontact printing, it is porous and could absorb solvents and small molecules which could change the concentrations of the drug in the droplets. In our case, we are quantifying the dosage in terms of fluorescence intensity after printing, which should circumvent this issue. For manufacturing purposes, other polymers such as photocurable perfluoropolyethers (PFPEs) or the thermoplastic polymer cyclic olefin copolymer (COC) may be more promising and further improvements of stamp materials can be applied [59]. Such a process would require replenishing of the inks on the stamps. Inkjet printing is a promising approach to high throughput production of the ink palettes [30]. Use of minimal amounts of reagents could be improved by acoustic droplet ejection technology [3].

## Stable encapsulation of small molecules into lipid droplet arrays

When organic droplets on a surface are immersed into solution, they are inevitably exposed to an air water interface, which can disrupt the droplets through dissolution and exposure to mechanical forces [60,61]. Fluid lipid multilayer patterns on surfaces can therefore be destroyed upon immersion into aqueous solution unless care is taken to prevent this [39,43,45,47,62,63]. Our previous works have involved careful immersion of phospholipid-based arrays in a low humidity environment [33,38,39,44,47,64,65]. Phospholipid droplet arrays are stable at low humidity for at least several weeks, but when exposed to high humidity the lipids can spread on the surface [66]. The triglyceride based formulations appear more stable when exposed to humidity although further studies are necessary to determine the extent of their stability for shipping purposes [38]. Faster immersion results in less exposure to the interface, and higher ink viscosity can be expected to stabilize the droplets upon immersion [61]. While other sources have improved immersion stability through freezing the droplets [45,47,66] and identification of presumably more viscous and less amphiphilic triglyceride-based formulations can allow more stable immersion [33,38,39,44,47,64,65]. When deformed droplets experience this immersion, they attempt to form a spherical cap shape in an effort to reduce interfacial tension [67]. The role this phenomenon plays is directly related to the stability of the emulsion itself. The interfacial energy present at the interface is determined by multiple factors including the presence and viscosity of a surfactant and the temperature of the droplets [38,68]. Previous studies have shown that using surfactant co-emulsifiers can help improve the stability of the droplets during immersion by lowering the amount of interfacial energy present in the system [38]. The use of castor oil/hexanoic acid in our formulations is to provide further stability to our droplet formulation. Other formulations have previously been shown to be stable in aqueous mediums for over 24 hours [69]. However, there are several time-limiting factors that should be considered when discussing stability and release of encapsulations including the pH and the temperature of the aqueous environment [69,70]. To further stabilize the arrays for stable immersion in ambient conditions, we have also tested a vapor-coating process [43,64,65]. In this process arrays are exposed to tetramethyl orthosilicate (TMOS) vapor, which appears to form a protective silicate layer that intercalates within the multilayers while still allowing the lipids to maintain fluidity. We found that process suitable for stabilizing the immersion of phospholipid-based lipid droplet arrays into aqueous solution [43]. Furthermore, cell adhesion of HeLa, MDCK, and HEK293 cells to the arrays was studied, with a sufficient number of these 3 cell types adhering to the arrays for drug screening assay [43]. Fig 4 shows new experiments designed to evaluate the ability for TMOS treatment to stabilize oil encapsulated hydrophobic and hydrophilic molecules for in vitro cellular delivery. A lipophilic fluorophore (rhodamine-PE) and a relatively water-soluble fluorophore (rhodamine B) were used as model small molecules to test for stable encapsulation. Two different lipid formulations were tested, one phospholipid based 1,2-dioleoyl-sn-glycero-3-phosphocholine (DOPC) and one triglyceride based (castor oil mixed with hexanoic acid). We found that this process can maintain stable encapsulation of both molecules after immersion into cell culture media for at least 90 minutes. Previous studies have shown that HeLa cells internalize the lipid droplets after about 1 hour in culture [39,44]. However, different assays that may require different encapsulation times would benefit from further characterization of the longer-term stability and release properties of these arrays. While it has been shown that some silicates can be toxic, silicates have been used in drug delivery [55]. It is conceivable that TMOS may react with some encapsulated drugs, possibly altering their bioactivity. That possibility should therefore be considered in future drug screens that make use of TMOS stabilization.

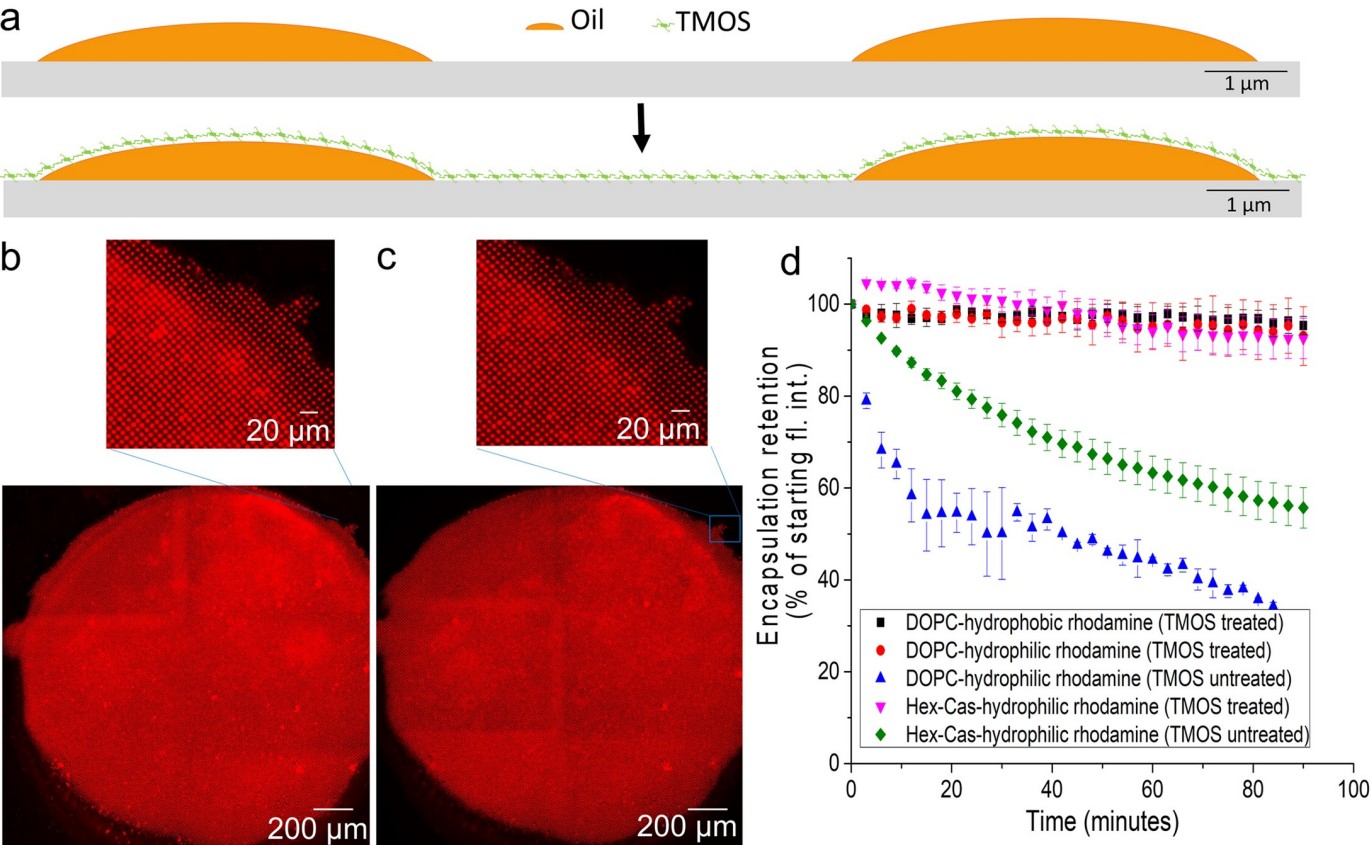

**Fig 4. Stable encapsulation using tetramethyl orthosilicate (TMOS) interaction with oils and phospholipids.** a) Schematic showing the top layer of silica-lipid complex of the TMOS treated lipids traps the hydrophilic molecules within the lipid droplet or multilayer and mitigates leakage and disruption during aqueous immersion. b) Fluorescence image of the water-soluble rhodamine B doped mixture of hexanoic acid / castor oil mixture after treatment with TMOS before immersion. At the top is a magnified section showing the subcellular nanostructures within the spot below. (c) Fluorescence image of the water-soluble rhodamine B doped mixture of hexanoic acid / castor oil mixture after treatment with TMOS and immersion under cell culture media for 1 hour. At the top is a magnified section showing the subcellular nanostructures within the spot below. (d) Graph showing a distinct improvement in the stability and resistance to leakage of various oil and phospholipid mixtures when treated with TMOS.

## Cellular uptake

Lipid droplets are used to confine reagents, having the drugs suspended in the oil mixture increases cellular uptake of the drug [30]. Lipid droplets with known concentrations are positioned onto a surface, the cells are grown on top of the array of droplets. Cells that don't migrate far are required, when they attach to the surface and over the droplets, the cells fuse with the droplets and adhere to the "empty" surface between droplets. Phospholipid encapsulated lipophilic drugs have been shown to only be taken up by cells that are in direct contact with the droplets [30]. With orthosilicate treated lipids, we expect a similar uptake mechanism based on endocytosis [30,35,36]. We found that cells are still able to internalize encapsulated materials from the arrays after the orthosilicate treatment process. Fig 5a and 5b shows confocal fluorescence data indicating uptake of the hydrophobic fluorophore rhodamine-PE and docetaxel by HeLa cells cultured on TMOS treated arrays. Cell nuclei are stained blue with DAPI, while the red signal indicates the localization of rhodamine-PE, and presumably the lipid carrier and encapsulated hydrophobic docetaxel. In this experiment we tested two different delivery vehicles, one phospholipid based, and one triglyceride based. Fig 5a shows the

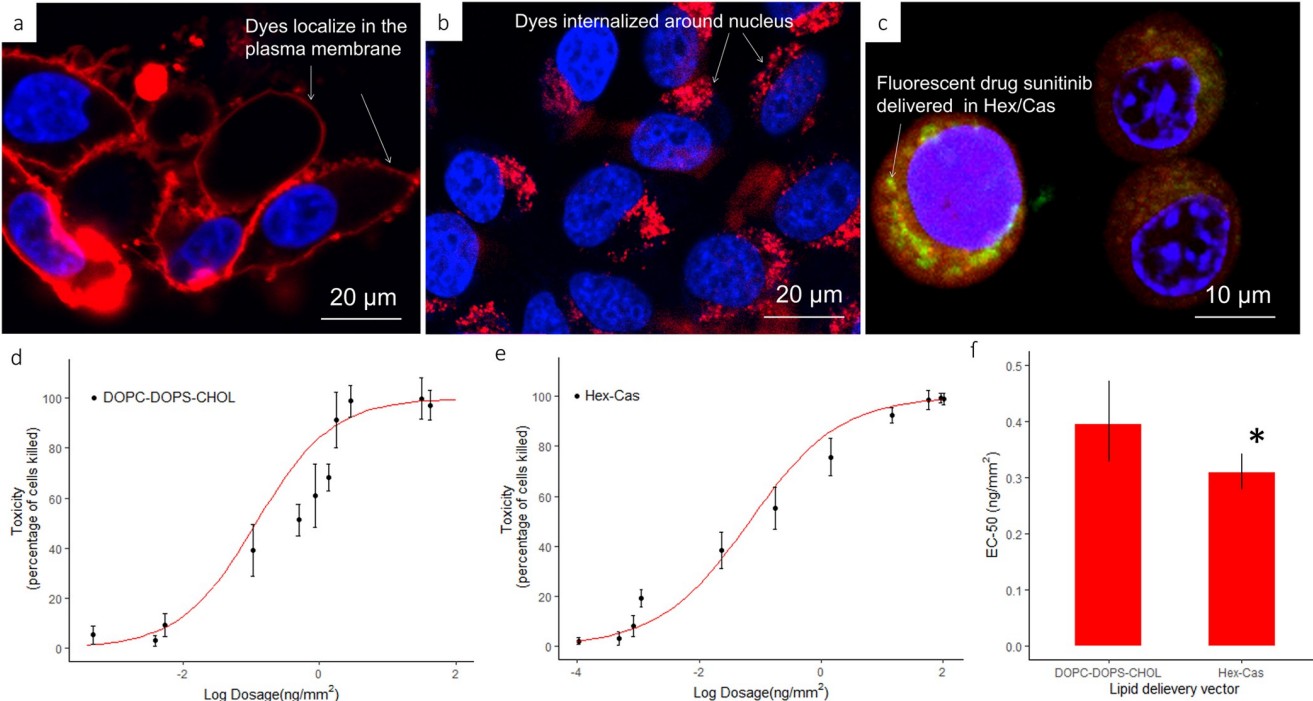

**Fig 5. Uptake of hydrophobic small molecules, including a fluorophore and docetaxel.** (a) Merged confocal fluorescence image of HeLa cells (blue DAPI stained nuclei) after delivery from a phospholipid-based microarray (DOPC / DOPS / cholesterol mixture). Rhodamine-PE is localized in the plasma membrane after 12 hours of incubation on array. (b) Merged confocal fluorescence image of HeLa cells after delivery from a trigliceride-based array (castor oil / hexanoic acid mixture). Rhodamine-PE is localized around the nucleus after 12 hours of incubation on array. (c) Merged confocal image showing fluorescent drug sunitinib (green) delivered to HeLa cells from on castor oil and hexanoic acid array. The drug is co-localized with the oil mixture around the nucleus. The image was taken 8 hours after incubation. (d) and (e) Graphs showing the range of toxic effect on HeLa cells when the DOPC / DOPS / cholesterol mixture is used vs the hexanoic acid / castor oil mixture. (f) Graph showing a significantly higher toxic effect on HeLa cells when the oil mixture is used as the delivery vector than when the DOPC / DOPS / cholesterol mixture is used.

results of the phospholipid [1,2-dioleoyl-sn-glycero-3-phosphocholine (DOPC) / 1,2-dioleoyl-sn-glycero-3-phospho-L-serine (DOPS) / cholesterol] based delivery, and Fig 5b shows the result of the triglyceride (castor oil / hexanoic acid) based delivery. The fluorophore delivered by the phospholipid-based vehicle can be seen to localize in the plasma membrane, while the triglyceride-based delivery resulted in localization around the nucleus. This shows that the intracellular destination of TMOS treated lipids depends on the delivery material used [71]. One delivery vehicle targets the membrane, while the other targets the cytoplasm. This is important as subcellular localization of drugs has been shown to be related to their activity [72].

As seen in Fig 5c, there is some variability in the uptake depending on the contact with and location and mobility of the cells on/around the dots when there is not full confluence. While the cell on the left shows an increased uptake amount compared to the other cells that also have green in them but at a lower and differing amount. Variation in cellular uptake is likely to depend on differences in cellular contact with the arrays, as well as heterogeneities within the cell population, such as differential stages of the cell cycle and gene expression. Furthermore, discrepancies in cell density can influence several different processes that could be considered vital for drug screening [73] with studies showing that cell plates of differing levels of confluence, show varied amounts of expression for specific proteins [74]. Depending on whether these proteins, serve cellular functions related to metabolism or are related to pathways

associated with the drug itself, there may be effects on the efficacy of the drug screen. The cell density also determines how many cells will be present on an individual spot. Depending on the variance and effect size, a sample size of approximately 50 cells per spot appears sufficient for two sample comparisons with the assays presented here [75]. A fluorescent drug, sunitinib, was delivered to the cells in a hexanoic acid / castor oil mixture (Fig 5c) to determine potency. A standard way to quantify the potency of a drug is to measure the effective concentration with a half maximal response, or EC-50. Lower EC-50 means less drug is needed to cause the same response. The range of toxic effect on HeLa cells when the oil mixture is used as the delivery vector than when the DOPC / DOPS / cholesterol mixture is used is shown in Fig 5d and 5e. A significantly higher potency was observed when the hexanoic acid / castor oil formulation was used as the delivery vehicle than when the DOPC / DOPS / cholesterol mixture was used (Fig 5f). While formulation is typically optimized after high throughput screening [47,76] due to limitations in throughput, improved delivery at the high throughput screening stage is a promising approach to repurposing or discovery of potentially therapeutic materials that are incompatible with traditional screening methods such as brick dust molecules [77–79].

Lipophilic drugs pose a challenge in drug screening, yet such compounds make up 40% of FDA approved oral drugs, and 75% of drug candidates under development [80,81]. While recent advances in the pharmaceutical industry have made progress in improving the bioavailability of water-soluble drugs, lipophilic based assays remains problematic [25,34,81]. For example, when a lipophilic drug is added to water, the actual amount of drug taken up by the cell, if any at all, is often unknown [44]. While microarrays have been used previously for drug delivery to cells, the cells take up the drugs from the array instead of through added solution and they are limited to water-soluble drugs [25,81]. While further formulations may be required to deliver varying drugs, our results indicate the suitability for lipid droplet microarrays to quantitatively deliver a lipophilic drug to different parts of a cell.

## Assay validation

Before conducting a screening experiment, it is important to validate the assay. This process typically involves testing of a positive control, or a compound that is known to give an efficacious response. We validated two assays, one for viral gene activation with applications in virology, and a second for culture of primary cancer cells with applications in functional precision medicine. An assay for viral activation of latently infected Kaposi Sarcoma-associated herpesvirus (KSHV) was validated using iSLK.219 cells using the water soluble compound doxycycline as the positive control [82]. Treatment of the cells with doxycycline induces viral lytic gene expression as a result of doxycycline-inducible expression of key viral switch gene RTA (Fig 6) [83]. High contrast was used for visibility. The cells include a constant GFP and an RFP that fluoresces when the lytic cycle is induced. Using the TMOS-treated, rhodamine-PE doped oil mixture (hexanoic acid / castor oil) containing the hydrophilic drug doxycycline and a control, the doxycycline successfully activated lytic viral gene expression as indicated by expression of RFP under the control of lytic gene promoter [83–85]. Cells cultured over lipid spots that didn't contain the drug (negative controls) did not express the reporter gene. As doxycycline is a water-soluble drug, with a calculated logP of -0.7 [82], its compatibility with lipid droplet microarray delivery in this assay indicates that this approach is not limited to lipophilic compounds. Further tests to determine the range of drug physicochemical properties compatible with this technique are therefore warranted.

Some cell lines require adhesion to the surface, specifically for many patient derived cell lines, so extra cellular matrix coatings are often used in primary cell culture. To verify a patient derived cell line could adhere to the lipid droplet microarrays with a surface coating and

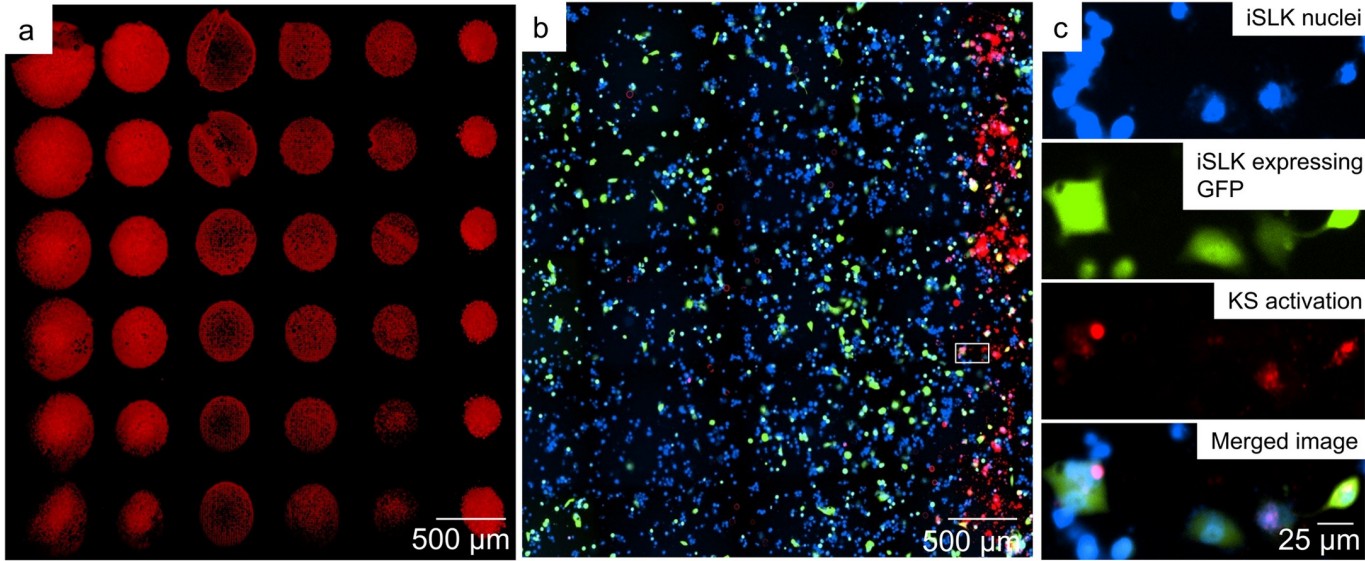

**Fig 6. Microarray uptake of a hydrophilic drug.** Virus assay In vitro delivery of hydrophilic drugs. (a) Doxycycline activates lytic viral gene expression in Kaposi Sarcoma-associated herpesvirus (KSHV) latently infected (iSLK.219) cells. Fluorescence image of TMOS-treated, rhodamine-PE doped oil mixture (hexanoic acid/castor oil) containing hydrophilic drugs. Rows are different hydrophilic drugs; columns are replicates of the same hydrophilic drugs. The column on the right is doxycycline while the other columns are negative controls. Image gamma corrected. (b) Fluorescence image of iSLK cells cultured over drug patterns for 48 hours. iSLK.219 cells constitutively express GFP; activated cells express RFP only over the doxycycline arrays. Cell nuclei are stained DAPI blue. (c) Zoom in on area in (b) indicated by white box. Cells fluorescing red due to RFP expression from viral activation. High contrast was used for visibility. All micrographs were taken with a 10x objective (a) and (b) are large images stitched together.

successfully deliver drugs, preliminary experiments were conducted with the patient derived cells, TM00199, maintained in PDX models (Fig 7). We then carried out assay validation experiments for anticancer drugs. We have previously shown by live cell imaging and live-dead cell staining that cell number remaining on the surface after washing to remove dead cells can be used as a proxy for cell viability [44]. Here, the lipid/drug arrays containing the anticancer drugs 1-(2-Chlorophenyl)-1-(4-chlorophenyl)-2,2-dichloroethane (CCDE) and Erlotinib were fabricated on collagen coated polystyrene with each dot representing one droplet (Fig 7a, 7c and 7f) [86,87]. Note that multiple droplets of sub-cellular dimensions contain the same drug within a certain area of the surface, as illustrated in Figs 1 and 2. Cells were cultured over the patterns and successfully attached to the surface (Fig 7b, 7d and 7g). Cells tend to aggregate over the lipid arrays, so the areas without lipid are lower in cell density, and the negative lipid control sometimes has the highest density, therefore the areas with only lipids were used as the negative control. Once the PDX-derived cell line TM00199 lung cancer PDCs adhered to the lipid droplet microarrays the delivery of drugs to the cells was also effectively demonstrated by both CCDE and Erlotinib having a significantly lower cell numbers than the pure lipid control (oils) (Fig 7h). The results suggest the suitability of lipid droplet microarrays for ex vivo testing of drug candidates on primary and patient derived cells for functional precision medicine applications [6–11].

## Screening

A preliminary test of the suitability for lipid droplet microarray technology to screen a lipophilic natural product library has been carried out as shown in Fig 8. Using the lipid droplet microarray, 92 different natural product extracts were screened for antiproliferative effects on

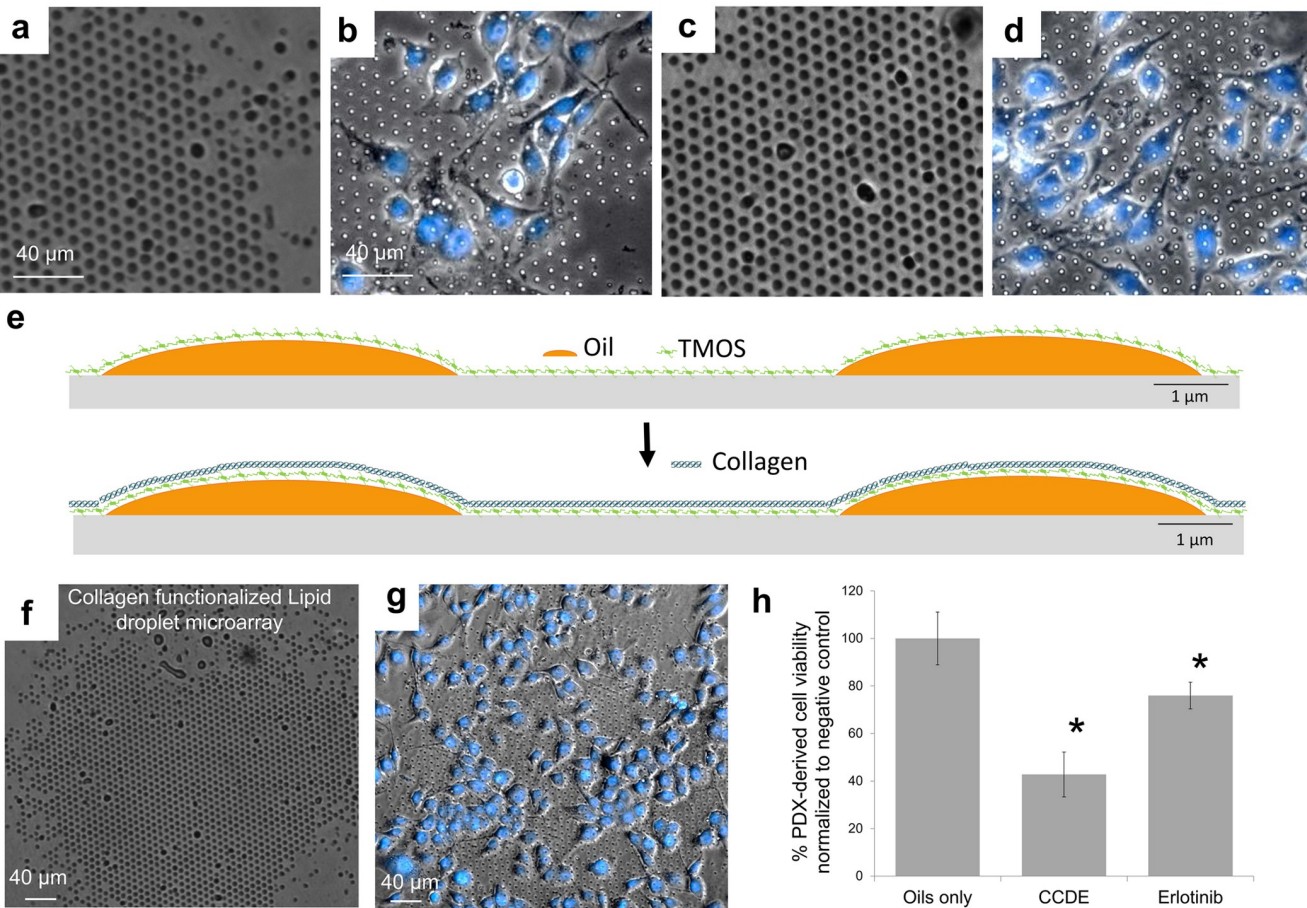

**Fig 7. Compatibility with extra cellular matrix coating for primary cell culture.** (a) Cell numbers of PDX-derived cell line TM00199 after 8 passages in vitro on lipid droplet microarrays. (a) and (c) are phase contrast images of lipid/drug arrays containing the drugs CCDE and Erlotinib, respectively, with each dot representing one droplet, fabricated on collagen coated polystyrene. (b) and (d) are cells cultured over the patterns in (a) and (c) respectively. Cells were washed three times vigorously to remove detached ones and the remaining adherent ones stained with DAPI, with DAPI stained nuclei in blue. The DAPI stained cells are counted as the viable cells over the selected area. Surface coating for adhesion of lung cancer PDCs to lipid droplet microarrays f. Lipid droplet microarray composed of a mixture of hexanoic acid and castor oil printed on collagen functionalized cell culture polystyrene. (g) Patient derived cells from PDX model line TM00199, passage #8, adhering to the arrays. (h) Cell count over the arrays was normalized to the pure lipid control (oils). *Denotes significantly difference from the control.

PDX-lung cancer cells. We used the most non-polar fractions of extracts from the US NCI Natural Products Repository library [88]. The hexanoic acid/castor oil and extract mixture was doped with rhodamine-PE, PDX-lung cancer cells were plated over the arrays and viability was calculated for each spot. (Fig 8a and 8b). Using this method, four hits were obtained with a view to scaling up the process to screen more extracts from the natural product library (Fig 8c). It is important to note that Z-scores in screening experiments are not typically used for hypothesis testing, but rather as a method to rank candidates in order to narrow down the possibilities for follow-up investigations. In this case, the low throughput screen is a first test of the technology, and higher throughput screens would have a stricter z-score threshold [89]. Fluid handling remains a limit in the throughput of this assay because the drugs must be added to the formulations prior to arraying. However, upon further scale up of the process described here, arrays can be mass produced from a single set of formulated plates. Since dosages on the order of nanograms per spot are needed, thousands of plates could be manufactured from milligrams of material.

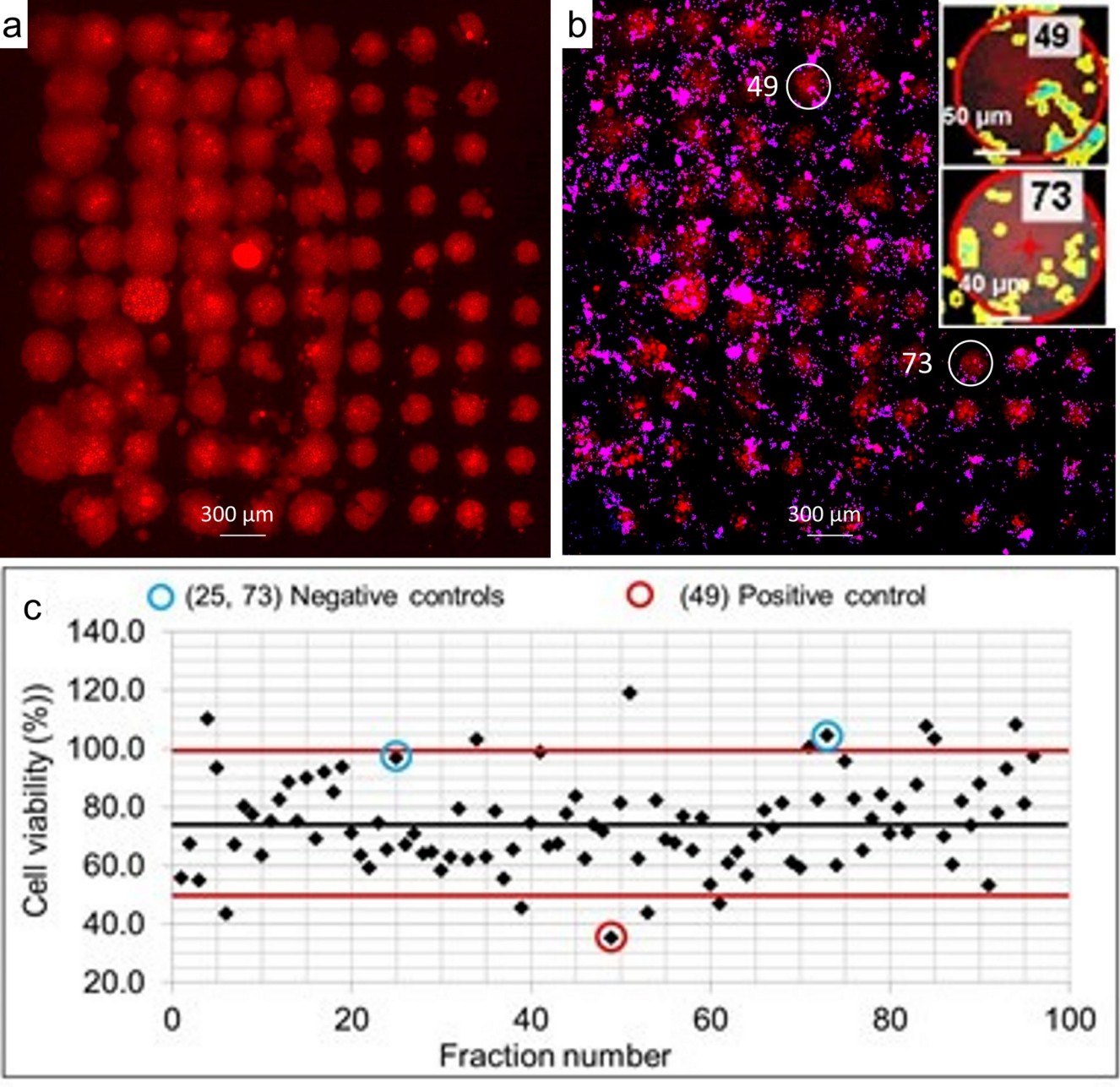

**Fig 8. Screening.** Lipid droplet microarray screening of 92 different natural product extracts for antiproliferative effects on PDX-lung cancer cells. a. Rhodamine-PE doped hexanoic acid/castor oil and extract mixture arrays. b. DAPI stained cells incubated over arrays in (a) for 24 hours. Insets show the cells over white circled areas. (73) is negative control with oil only, (49) is positive control of the anticancer drug erlotinib hydrochloride. c. Graph of cell viability over each of the fractions screened. Grey line indicates mean, viabilities of +/-1.5 SD are indicated by the red lines as hits in this low throughput screen. Fractions above the top red line are likely to promote cell growth.

## Conclusions

We have presented a scalable microarray screening process that is compatible with compounds of varying physicochemical properties. First a scalable printing nanointaglio printing process has been demonstrated for arrays with controllable dosages. A vapor coating process has been

demonstrated that allows for stable immersion into aqueous solution. These arrays can encapsulate lipophilic and water-soluble drugs. Two assays were validated and a preliminary screen of 92 different natural products was conducted. In addition to lowering the cost of high throughput screening and making it accessible to labs, the arrays have potential to overcome limits with current screening technology. The resulting technology has potential for use in BSL 3/4 containment, for use on fresh primary cells from patients, and for generally increasing throughput while decreasing cost of high throughput screening.

## Materials and methods

### Array fabrication

1,2-dioleoyl-sn-glycero-3-phosphocholine (DOPC) and 1,2-dimyristoyl-sn-glycero-3-phosphoethanolamine-N- (lissamine rhodamine-B-sulfonyl, ammonium salt (rhodamine-PE) and 1,2-dioleoyl-sn-glycero-3-phospho-L-serine (DOPS) dissolved in chloroform were purchased from Avanti Polar Lipids, aliquoted into a glass vial and dried in a vacuum. Rhodamine-PE was used to dope the DOPC for fluorescence visualization and characterization. Deionized water was added to the vials containing the dried lipids to form liposomes. The samples were sonicated for 10 minutes and aliquoted into microtiter plates. The liposomes were then microarrayed using an Arrayit SpotBot pinspotter, onto a flat polydimethylsiloxane (PDMS) pallet and dried in a vacuum for 2 hours. For the intracellular localization of lipids experiment formulations of rhodamine-PE doped DOPC / DOPS / cholesterol and Hexanoic Acid / Castor oil were printed and stamped onto glass bottom dishes.

### Nanointaglio

For hand stamping method, a polydimethylsiloxane (PDMS) stamp with well features of 5 μm diameter and 2.5 μm depth was inked by pressing the patterned surface onto the ink palette. The inked stamp was then pressed onto polystyrene or glass cell culture surface with the thumb to obtain spots made up of smaller subcellular nanostructures with ~5 μm lateral dimensions.

For the vertical stamping method, a vertical press system was designed to be used for weight dependent stamping of the arrays. The vertical press system involves a top-down press system using controlled pressure for small well plates. A 1 cm thick PDMS stamp with well features of 5 μm diameter and 2.5 μm depth on one surface were fabricated for vertical printing and inked with the array by gently pressing the surface with the wells over the array on the PDMS pallet. The inked stamp was then affixed to a flat rigid plastic backing. The stamp was then pressed to the well plate culture surface. Weights generating forces ranging from 1N-16N were applied to the stamp. For the rolling press method, a 3mm thick PDMS stamp with well features of 5 μm diameter and 2.5 μm depth on one surface were fabricated for rolling press printing. The PDMS was affixed to a roller with the side with the well features facing outward. The stamped was inked with the arrays by gently rolling the affixed stamp over the microarrayed pallet making sure the stamps were aligned to capture all the array spots. The inked roller stamp was then gently rolled over the cell culture glass surface to produce the nanointaglio arrays.

### Optical microscopy

Epifluorescence microscopy was done using a Ti-E inverted microscope (Nikon Instruments, Melville, NY, USA) fitted with a Retiga SRV (QImaging, Surrey, BC, Canada) CCD camera (1.4 MP, Peltier cooled to −45˚C). Rhodamine-PE doped lipid structures were imaged using the G-2E/C filter. DAPI was imaged using the UV-2E/C fluorescence filter, and GFP were

imaged using the B-2E/C. Confocal microscopy was conducted on an Olympus FV1000 Confocal Laser Scanning Biological Microscope. A 60x oil objective and a 405-nm diode laser (for DAPI) or a 543-nm He/Ne laser (for rhodamine) was used. The fabricated arrays were characterized using optical fluorescence microscopy. For the 92-extract screen we used a single dosage of 1.2 pg/mm$^2$ +/- 20% as determined by fluorescence calibration.

## Atomic force microscopy

AFM heights of the lipid prints were measured in tapping mode with a Dimension Icon AFM (Bruker, Billerica, MA, USA) and tapping mode AFM cantilevers (FESPA, 8 nm nominal tip radius, 10–15 μm tip height, 2.8 N/m spring constant, Bruker, Billerica, MA, USA).

## Scanning electron microscopy

Samples for scanning electron microscopy (SEM) photos were prepared several days prior to imaging which was done using the FEI Helios G4 UC located at the National High Magnetic Field Laboratory. Samples that were on glass coverslips were placed on silicone holders and fastened using carbon tape. These samples were then coated with a 2 nm thick layer of platinum using 30 mA of current in a Leica EM ACE600 sputter coater prior to their placement in the SEM vacuum chamber. Several regions of interest were identified using a lower magnification (100x) and micrographs were taken of the samples with increasing magnification.

## Contact angle measurements

Droplets with volumes between 100–150 μL of distilled water were placed onto surfaces of interest using a pipette. Photographs were taken of the drops with volumes adjusted for advancing and receding contact angle measurements parallel to the surface using a cell phone camera and white backlight. Contact angles were determined from the images using an open-source contact angle plugin designed by Marco Brugnara for use in ImageJ. Each experiment was repeated 10 times.

## Dye encapsulation and leakage

Two dyes, hydrophilic rhodamine B and hydrophobic rhodamine-PE, were each mixed separately with DOPC and with a mixture of castor oil and hexanoic acid and used for the fabrication of nanointaglio arrays. In at least 4 array samples, each mixture of DOPC/rhodamine B, DOPC/rhodamine-PE, hexanoic acid / castor oil /rhodamine B and hexanoic acid / castor oil /rhodamine-PE were fabricated. Half the samples were treated with TMOS, and the other half were left untreated. Both TMOS treated and untreated microarrays were immersed under cell culture media and a fluorescence microscopy time lapse taken to record the change in fluorescence intensity over the period of 90 minutes. The relative change in fluorescence intensity as a percent of the starting intensity was plotted as a measure of dye leakage.

## Orthosilicate treatment

Two glass vials, one containing 2mL of tetramethyl orthosilicate, purchased from Sigma Aldrich$^®$ and the other containing deionized water were placed into a larger sealable glass container. The lipid multilayers were placed in the larger glass container with the other two vials. The container was then sealed tight and left at room temperature for 4 hours after which the printed lipids were removed and ready for cell culture.

## Collagen coating

To ensure proper coating, 200 µL of 0.01% collagen was aliquoted onto culture substrate and incubated for 2 hours at 37˚C. Collagen solution was aspirated off culture substrate and surface washed 3X in HBSS buffer. The substrate was air dried under sterile conditions for 1 hour before the lipid array was then stamped onto the collagen coated surface. Some arrays were treated with TMOS vapor for 4 hours. TMOS treated arrays were immersed under another 200 µL of 0.01 collagen solution for 2 hours at 37˚C. Collagen solution was aspirated off the array and washed 3X with HBSS buffer.

## PLL coating

Poly-l-lysine purchased from Sigma (P6407-5MG) was and used to coat slides to promote cell adhesion. The poly-l-lysine was thawed to room temperature and a 1:10 dilution was made in deionized water; the slides were placed inside solution for 5 minutes and left to dry in the oven at 60˚C for one hour.

## Cell culture

The cell types used included HeLa cells purchased from ATCC. iSLK.219 cells were obtained from Jinjong Myoung and cultured as previously described [84,85]. PDX cells used in these experiments were cultured from PDX lung tumours, TM 0199 (Jackson Labs). These cells express EGFR L858R mutations. For the intracellular localization of lipids experiment the cells were incubated for 6 hours, stained with DAPI, and then imaged using confocal microscopy to determine the intracellular location of the lipids. Microarrays were immersed under cell culture media without cells. The cells were then added onto the already immersed arrays. Cells were incubated over arrays for the times shown below.

## Assay for drug efficacy

For EC50 determinations, the lipid/drug mixtures were stamped multiple times into cell culture wells to obtain the variation in quantities (heights) lipid/drugs deposited. The heights were converted to dosages using the calibration method described previously [90,91]. Cells were cultured for 24 hours in all tests. Cells were washed three times vigorously to remove detached ones and the remaining adherent ones stained with DAPI. The DAPI stained cells are counted as the viable cells over the selected area. Dose response curves were plotted in Origin and EC50 values extracted from the plots. Toxicity experiments were done in triplicate with 3 repeats and error bars represent standard deviation from the nine experiments. The arrays were fluorescently imaged using the Texas red filter of the Nikon Eclipse Ti microscope. For optical calibration purposes the images were taken, and fluorescence intensities images were taken with an exposure time ranging from 2 microseconds to 2 seconds. AFM measurements of the imaged fluorescent docetaxel lipid multilayers were taken. Calibration was done as previously described [46].

## Natural product extracts

92 natural product extract fractions dissolved in DMSO obtained from the NCI at Frederick National Laboratory for Cancer Research were mixed with 1,2-dimyristoyl-sn-glycero-3-phosphoethanolamine-N- (lissamine rhodamine-B-sulfonyl) (ammonium salt) were used [50]. 92 of the most non-polar fractions of extracts (Acetone (ACN) / dichloromethane (DCM), and methanol (MeOH) / DCM fractions) were mixed with (rhodamine-PE) doped oil mixture (castor oil and hexanoic acid) in a 384 microtiter well plate. Three control wells, two negative

controls with only DMSO and a positive control of Erlotinib, were added to the microwells for micro arraying. Mixing of extracts/drugs with oils was done by pipetting up and down. The microtiter plate was then placed in a SpotBot® Extreme pinspotter microarrayer for printing. A 10X10 array design was used to print the array onto a 1mm thick polydimethylsiloxane PDMS pallet. Dwell time of 1 s was used for printing. The printed array pallet was placed in a vacuum overnight for drying out the DMSO before nanointaglio stamping. Mixing of other drugs (CCDE, doxycycline) with oils for printing was done in the same fashion. The most non-polar fractions of extracts from the US NCI Natural Products Repository library were used. Acetone (ACN) / dichloromethane (DCM), and methanol (MeOH) / DCM fractions were mixed with the oil carrier (hexanoic acid / castor oil mixture in 1:1 ratio), doped with rhodamine-PE and used for the screen. We performed the screen at a single dosage of 1.2 pg/mm2 +/- 20% as determined by fluorescence calibration. PDX-lung cancer cells were plated over the arrays for 24 hours, washed, stained with DAPI, and then counted. Viability was calculated for each spot and significance assigned to values more than 1.5 standard deviation above or below the population mean (74.5). Using the lipid droplet microarray 92 natural product extracts were screened, with two negative controls and a positive control (Fig 7). With a negative control set to a cell viability of 100 percent and a hit was considered if +/-1.5 standard deviations. Four hits were attributed to this screen.

## Supporting information

**S1 File.**
(DOCX)

## Acknowledgments

The US NCI Natural Products Repository library was kindly provided by Barry O'Keefe at the Frederick National Laboratory for Cancer Research. The authors thank Troy W. Lowry, James Penate, Cecelia Bouaichi, and Parker Pecina, for help with the array fabrication and Javier Mariscal and Juan Reza for reviewing the paper.

## Author Contributions

**Conceptualization:** Steven Lenhert.

**Data curation:** Tracey N. Bell, Aubrey E. Kusi-Appiah, Vincent Tocci, Lei Zhu.

**Formal analysis:** Vincent Tocci, Pengfei Lyu, Hongyuan Cao.

**Funding acquisition:** Steven Lenhert.

**Investigation:** Aubrey E. Kusi-Appiah.

**Methodology:** Aubrey E. Kusi-Appiah, Fanxiu Zhu, David Van Winkle, Mandip S. Singh, Steven Lenhert.

**Resources:** Lei Zhu, Fanxiu Zhu, Mandip S. Singh.

**Supervision:** David Van Winkle, Hongyuan Cao, Steven Lenhert.

**Writing – original draft:** Tracey N. Bell, Steven Lenhert.

**Writing – review & editing:** Tracey N. Bell, Aubrey E. Kusi-Appiah, Vincent Tocci, Steven Lenhert.

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
