## [Decision Letter · Decision Letter 0]

31 Jan 2024

PONE-D-24-00790Scalable lipid droplet microarray fabrication, validation, and screeningPLOS ONE

Dear Dr. Lenhert,

Thank you for submitting your manuscript to PLOS ONE. After careful consideration, we feel that it has merit but does not fully meet PLOS ONE’s publication criteria as it currently stands. Therefore, we invite you to submit a revised version of the manuscript that addresses the points raised during the review process.

We look forward to receiving your revised manuscript.

Kind regards,

Dr.Gufran Ahmad

Academic Editor

PLOS ONE

Journal Requirements:

10.1016/j.nano.2009.04.002

In your revision ensure you cite all your sources (including your own works), and quote or rephrase any duplicated text outside the methods section. Further consideration is dependent on these concerns being addressed

4. Please expand the acronym “NIH” (as indicated in your financial disclosure) so that it states the name of your funders in full.

"NIH R01 GM107172"    

6. We noted in your submission details that a portion of your manuscript may have been presented or published elsewhere. "A previous version of this paper has been uploaded to the following preprint archive:

arXiv:2210.07377v1 Available from doi.org/10.48550/arXiv.2210.07377 (Preprint)." Please clarify whether this [conference proceeding or publication] was peer-reviewed and formally published. If this work was previously peer-reviewed and published, in the cover letter please provide the reason that this work does not constitute dual publication and should be included in the current manuscript.

7. When completing the data availability statement of the submission form, you indicated that you will make your data available on acceptance. We strongly recommend all authors decide on a data sharing plan before acceptance, as the process can be lengthy and hold up publication timelines. Please note that, though access restrictions are acceptable now, your entire data will need to be made freely accessible if your manuscript is accepted for publication. This policy applies to all data except where public deposition would breach compliance with the protocol approved by your research ethics board. If you are unable to adhere to our open data policy, please kindly revise your statement to explain your reasoning and we will seek the editor's input on an exemption. Please be assured that, once you have provided your new statement, the assessment of your exemption will not hold up the peer review process.

Additional Editor Comments :

Thank you for submitting your manuscript to PLOS ONE. After careful consideration, we feel that the work is timely, and the manuscript has merit but does not fully meet PLOS ONE’s publication criteria as it currently stands. Therefore, we invite you to submit a revised version of the manuscript that could address the points raised by all the reviewers. The manuscript requires a major revision prior to a final decision.

Reviewers' comments:

Reviewer's Responses to Questions

**Comments to the Author**

1. Is the manuscript technically sound, and do the data support the conclusions?

Reviewer #1: Yes

Reviewer #2: Yes

2. Has the statistical analysis been performed appropriately and rigorously? 

Reviewer #1: Yes

Reviewer #2: Yes

3. Have the authors made all data underlying the findings in their manuscript fully available?

Reviewer #1: Yes

Reviewer #2: Yes

4. Is the manuscript presented in an intelligible fashion and written in standard English?

Reviewer #1: Yes

Reviewer #2: Yes

5. Review Comments to the Author

Reviewer #1: In this manuscript, authors have fabricated liquid droplet microarray for cellular validation and screening using nanointaglio. And, the authors have succeeded to screen the screening. From these achievements, this manuscript seems useful for high-throughput screening for cell analysis. However for publication of this manuscript, authors should reconsider the several points.

1. The stability and lifetime of droplets. Is these droplets can be printed on the substrate stably? If possible, the life time also should be discussed.

2. Are the cell density affect to the screening efficiency?

3. If possible, after the printing with different pressures, differences of sizes of droplet should be indicated using SEM imaging.

Reviewer #2: The work reported in the manuscript is interesting in terms of both the method to prepare droplets, and the cell responses to droplets. The former is the focus of the manuscript. In forming the droplets, the first step is quite ordinary, but it is clever to split the ink droplets by using intaglio stamp. I have the following comments.

1. Figure 2 may better show the images of the droplet array prepared under the optimal pressure of 16 N/m^2, along with current representative images. It is important to show 100% uniformity droplets.

2. What is the interfacial tension between aqueous medium and the lipid droplets? This tension determines whether the deformed droplets can readily resume a spherical cap shape.

3. The authors pointed out that the droplets are not stable as they are immersed into the aqueous medium. The reason for such instability is the development of four phase contact line, which is unavoidable in the process of bringing oil-like droplets into water, or vice verse. See the detailed mechanism: Splitting droplets through coalescence of two different three-phase contact lines. Soft matter 15 (30), 6055-6061 (2019). The approach taken by the authors to improve the stability of the droplets is ok. An alternative way may be to improve the viscosity of the droplets, or control the speed for immersion. The authors may provide these possibilities in their outlook for future work.

4. The surface gettability of the stamps would play an important role in precisely distributing the droplet liquid. What are the advancing and receding contact angles of the stamp materials?

6. PLOS authors have the option to publish the peer review history of their article (what does this mean?). If published, this will include your full peer review and any attached files.

Reviewer #1: No

Reviewer #2: **Yes: **Xuehua Zhang

---

## [Author Response · Author response to Decision Letter 0]

18 Mar 2024

Please find a detailed response to reviewer document enclosed.

---

## [Decision Letter · Decision Letter 1]

17 May 2024

Scalable lipid droplet microarray fabrication, validation, and screening

PONE-D-24-00790R1

Dear Dr. Lenhert,

We’re pleased to inform you that your manuscript has been judged scientifically suitable for publication and will be formally accepted for publication once it meets all outstanding technical requirements.

Kind regards,

Jian Xu, Ph.D.

Academic Editor

PLOS ONE

Additional Editor Comments (optional):

Reviewers' comments:

Reviewer's Responses to Questions

**Comments to the Author**

1. If the authors have adequately addressed your comments raised in a previous round of review and you feel that this manuscript is now acceptable for publication, you may indicate that here to bypass the “Comments to the Author” section, enter your conflict of interest statement in the “Confidential to Editor” section, and submit your "Accept" recommendation.

Reviewer #1: All comments have been addressed

Reviewer #2: All comments have been addressed

2. Is the manuscript technically sound, and do the data support the conclusions?

Reviewer #1: Yes

Reviewer #2: Yes

3. Has the statistical analysis been performed appropriately and rigorously? 

Reviewer #1: Yes

Reviewer #2: Yes

4. Have the authors made all data underlying the findings in their manuscript fully available?

Reviewer #1: Yes

Reviewer #2: Yes

5. Is the manuscript presented in an intelligible fashion and written in standard English?

Reviewer #1: Yes

Reviewer #2: Yes

6. Review Comments to the Author

Reviewer #1: The authors were able to accurately reconsider the points raised by the reviewers. Revisions are satisfactory. Hence, this manuscript is able to publish in PLOS ONE.

Reviewer #2: The authors have made careful revision. All the comments haven been addressed adequately. I have no more comments.

7. PLOS authors have the option to publish the peer review history of their article (what does this mean?). If published, this will include your full peer review and any attached files.

Reviewer #1: **Yes: **Tatsuro Endo

Reviewer #2: No

---

## [Editor Report · Acceptance letter]

27 Jun 2024

PONE-D-24-00790R1 

PLOS ONE

Dear Dr. Lenhert, 

I'm pleased to inform you that your manuscript has been deemed suitable for publication in PLOS ONE. Congratulations! Your manuscript is now being handed over to our production team.

Kind regards, 

on behalf of

Dr. Jian Xu 

Academic Editor

PLOS ONE